# Visibility Graph Feature Model of Vibration Signals: A Novel Bearing Fault Diagnosis Approach

**DOI:** 10.3390/ma11112262

**Published:** 2018-11-13

**Authors:** Zhe Zhang, Yong Qin, Limin Jia, Xin’an Chen

**Affiliations:** 1State Key Lab of Rail Traffic Control and Safety, Beijing Jiaotong University, Beijing 100044, China; zhangzhe@bjtu.edu.cn (Z.Z.); 15114217@bjtu.edu.cn (X.C.); 2Beijing Research Center of Urban Traffic Information Sensing and Service Technologies, Beijing Jiaotong University, Beijing 100044, China

**Keywords:** rolling bearing, nonlinear vibration signals, visibility graph features, Gaussian Markov random fields

## Abstract

Reliable fault diagnosis of rolling bearings is an important issue for the normal operation of many rotating machines. Information about the structure dynamics is always hidden in the vibration response of the bearings, and it is often very difficult to extract them correctly due to the nonlinear/chaotic nature of the vibration signal. This paper proposes a new feature extraction model of vibration signals for bearing fault diagnosis by employing a recently-developed concept in graph theory, the visibility graph (VG). The VG approach is used to convert the vibration signals into a binary matrix. We extract 15 VG features from the binary matrix by using the network analysis and image processing methods. The three global VG features are proposed based on the complex network theory to describe the global characteristics of the binary matrix. The 12 local VG features are proposed based on the texture analysis method of images, Gaussian Markov random fields, to describe the local characteristics of the binary matrix. The feature selection algorithm is applied to select the VG feature subsets with the best performance. Experimental results are shown for the Case Western Reserve University Bearing Data. The efficiency of the visibility graph feature model is verified by the higher diagnosis accuracy compared to the statistical and wavelet package feature model. The VG features can be used to recognize the fault of rolling bearings under variable working conditions.

## 1. Introduction

Nowadays, rotating machines are used widely in industrial systems, and they have become the most critical equipment in many industrial systems. In the main components of rotating machines, the fault of rolling bearings is the most frequent reason for unexpected machine breakdown and results in economic loss [1]. Therefore, the material control in the quality design process [2,3] and the fault diagnosis of rolling bearings in the operation process have attracted considerable attention of engineers and scholars in order to reduce the system breakdown. This paper focuses on the fault diagnosis study of rolling bearings in the operation process.

Several vibration and acoustic measurement-based methods have been developed for the detection of defects in rolling element bearings. To the best of our knowledge, the vibration-based diagnosis method is the most widely employed because vibration signals contain a wealth of information about the structure dynamics [4,5,6]. The most common steps of fault diagnosis are: collect the vibration signals using sensors, extract the fault features through a signal processing method [7] and finally detect or recognize the fault classes through classifiers [1]. In order to get accurate diagnosis results and develop a more efficient diagnosis method, many feature models have been developed. For example, the statistical feature model can produce a time-domain and frequency-domain statistical feature set, and these features are exclusively used to recognize faults of bearings and other machines [8,9]. Complex envelope analysis and wavelet packet analysis are used to get the envelope features and wavelet packet features [10,11,12,13]. In recent years, the image features have also been extracted from the vibration spectrum to detect the faults of bearings [1,14].

Researchers often overlook the feature classification and arbitrarily choose the number of features [15]. However, it is not certain that all the features contribute to fault diagnosis. Further, the irrelevant features increase the measurement and storage requirements, increase computing cost and may lead to the poor prediction performance of classifiers [16]. Therefore, the necessary processing step before fault diagnosis is to select good features through feature selection (FS). Usually, the relevant features are selected and the redundant features are eliminated after FS [17]. A survey on the FS can be found in [18,19].

Artificial intelligence (AI) has been developed and applied to reduce the breakdowns of manufacturing systems [20,21]. Researchers prefer to use machine learning-based classifiers to estimate the classification performance of their proposed features, even though some works only use visual inspection of peaks in frequency graphs for fault diagnosis [22]. The artificial neural network (ANN) was developed to recognize the fault classes of rolling bearings, and the time-domain features were used as the input of the ANN classifier in [23]. However, the various loading conditions make the ANN task very complicated. The support vector machine (SVM) is another classifier widely used in the fault diagnosis of rolling bearings. SVM classifies better than ANN because of the principle of risk minimization [24]. The inputs of SVM used in previous work are the intrinsic mode function envelop spectrum, time- and frequency-domain features [25,26].

In sum, feature extraction and selection are very important for fault diagnosis of rolling bearings because the classification performance depends on the quality of the dataset and largely the features used. Although many feature models have been developed, it is still very difficult to extract them correctly due to the nonlinear/chaotic nature of the vibration signal, so the number of features is growing now in order to recognize the fault or estimate the health of rolling bearings more precisely and efficiently [27].

It is well known that the vibration signals are one kind of time series data acquired by sensors. Recently, a new and simple method was introduced to convert a time series to a network, called the visibility graph (VG). It has been proven that the network structure resulting from the VG approach inherits some of the properties of the time series data [28]; for example, periodic series resulting in regular graphs, random series resulting in random graphs and fractal series converting into scale-free networks [29]. The visibility graph concept comes from the space and geometry theory; each node of the graph represents a location in a special space, and the edge between two nodes shows that the two nodes can see each other [30]. For the time series data, the nodes represent the data values in a planar coordinate system, and the edge between two nodes shows that the two data values can see each other. After the VG was proposed, researchers presented improved versions of the VG approach such as the horizontal visibility graph (HVG) [31], the limited penetrable visibility graph (LPVG) [32,33] and the parametric natural visibility graph (PNVG) [34]. The VG approach has been studied increasingly in many fields such as economic/marketing [35,36], ecology [37] and health analysis [38,39]. However, the VG approach has not been used in fault diagnosis of mechanical components such as gear and bearings. Furthermore, it has been shown that the VG approach has a noise resistance ability [40]. Therefore, we investigate whether VG can be used as an effective tool for diagnosing rolling bearing faults in this paper.

The graph analysis method has been applied in many areas including traffic systems [41], biology [42] and communication [43], except vibration signal processing. Using the VG approach, we can obtain a graph that has special topological properties. It allows us to use the complex graph approach to analyze the vibration signals. Fortunately, the features or properties of the complex graph have been discussed in previous studies. The features to measure the hidden information in the graph include graph density, degree distribution [44], average path length [45], graph diameter [46], the clustering coefficient, etc. [47].

However, there is little research to bridge graph analysis and signal-based fault diagnosis, and the graph analysis method is also limited. In this paper, the VG approach is used successfully to transform the vibration signals into a graph or a binary matrix. The global VG features are proposed based on the graph analysis, and the local VG features are proposed based on the texture analysis method of images. Therefore, a new feature model of vibration signals is formulated in this paper.

In this paper, we present a novel nonlinear feature model that can extract the visibility graph (VG) features from the time series vibration signals. A novel signal processing method is proposed. The vibration signals of rolling bearings are converted into a binary matrix. The vibration signals are firstly analyzed from the viewpoint of graph and image processing. A new set of VG features is extracted from the vibration signals. The sequential feature selection (SFS) algorithm is applied to select the VG feature subset that can realize the best diagnosis performance. The artificial neural graph (ANN), K-nearest neighbor (KNN) [48,49] and support vector machine (SVM) are used as classifiers for the VG features. The effectiveness of the selected VG features in the fault diagnosis of the rolling bearings is verified by comparison with the statistical feature model and wavelet package feature model. Because the working condition and the fault level are also considered in the experiments, the VG features can be used to recognize the fault class and the corresponding fault level of rolling bearings under variable working conditions.

To motivate this work, the Case Western Reserve University (CWRU) Bearing Data [22] are used in the experiments. The study begins with presenting the methodological framework of VG feature selection in Section 2. According to the framework, firstly, the method to convert the data into a matrix is proposed in Section 3; secondly, the VG feature extraction and selection method is proposed in Section 4. An experiment is presented in Section 5 in order to draw some conclusions in Section 6.

## 2. Methodological Framework

The framework to perform the bearing fault diagnosis in this paper is illustrated in Figure 1. The sequence of signal processing steps is inspired by [50] and can be described in sequence as follows:Data segmentation: The signals are segmented according to the sample rate and rough shaft speed to ensure each obtained sample covers several circles of signals.Feature extraction: The visibility graph method is used to convert the acceleration signals into a binary matrix. From the matrix obtained, considering the feature model of the complex graph and image, we extract the feature vector within this model. We do this for all available VG feature models and produce the VG feature pool.Feature selection: We select the optimal number of VG features based on the diagnosis performance. Multiple classifiers can be used to estimate the performance of features.Results analysis: The advantage of the VG feature model is validated by comparing the performance of the VG feature model, the statistical feature model and the wavelet package feature model.

## 3. Visibility Graph Construction

We denote the vibration signals by s(i), where i=1,2,…,N is a discrete time step indexing the time of collecting each signal. In order to process the data using our method, we have applied the visibility graph approach for mapping vibration signals in a corresponding complex graph in the experiments. The visibility graph method has been investigated in previous studies; however, the application of this approach is bounded. The visibility algorithm is a map that assigns each signal point to a node/vertex in a complex graph. Two nodes will be connected whenever one can draw a line in the time series space without intersecting any intermediate signal height. For the vibration signals, two signals (*i*, s(i)) and (*j*, s(j)) will have visibility and consequently will become two connected nodes of the associated graph, if any other data (*k*, s(k)) placed between them fulfill:(1)$$s\left(k\right)<s\left(j\right)+\left(s\left(i\right)-s\left(j\right)\right)\frac{i-k}{j-i}$$

The visibility graph can reflect the characteristics of the signal envelope. If the two signals can be connected according to Equation (Equation 1), the envelop between them is concave, otherwise the envelop is convex because there are some larger signals between them. Figure 2 depicts the relationship between the envelope and visibility graph. Signal $s_{1}$ cannot be connected to signal $s_{2}$, and signal $s_{3}$ cannot be connected to signal $s_{4}$ (dashed lines) because the signals under the convex envelope curves (EC) obstruct the visibility between them. Signal $s_{2}$ can be connected to signal $s_{3}$, and signal $s_{4}$ can be connected to signal $s_{5}$ (solid lines) because the signals under the concave envelope curves cannot obstruct the visibility between them.

Therefore, we can use Jensen’s inequality [51] to describe the relationship between the visibility graph and the signal envelop. Let s(t) be the signal envelope connecting signals (*i*, s(i)) and (*j*, s(j)). The two signals can be connected if s(αi+(1−α)j)<αs(i)+(1−α)s(j) for any α∈[0,1]. Using the VG algorithm, the acceleration signal is mapped into an ordered graph with a special spatial structure.

The obtained visibility graph can be represented by its adjacent matrix *M* (symmetric matrix), whose elements $m_{i,j}=0$ or 1. If the signals $s_{i}$ and $s_{j}$ can be connected according to Equation (Equation 1), $m_{i,j}=1$, otherwise, $m_{i,j}=0$, that is:(2)$$m_{i,j}=\left\{\begin{array}{cc} 1, & s\left(k\right)<s\left(j\right)+\left(s\left(i\right)-s\left(j\right)\right)\frac{i-k}{j-i},i<k<j\\ 0, & \mathrm{otherwise}. \end{array}\right.$$

Based on the obtained adjacent matrix of the visibility graph (VGAM), firstly, we can extract visibility graph features from VGAM, for example graph density and graph index complexity; however, the graph features prefer to describe the global characteristics of VGAM rather than local properties. Meanwhile, the VGAM can also be seen as binary images (Figure 3), which only include zero and one. Therefore, the local properties of VGAM can be extracted by using the image processing methods. The texture features of VGAM resulting from the GMRF method are extracted in this paper because the GMRF method performs better than other methods without considering the rotated invariant [52]. Therefore, the VG features including global and local features of VGAM can be extracted for bearing fault diagnosis.

## 4. VG Features’ Extraction and Selection

In this section, the candidate VG features and the feature selection algorithm are described. We can construct VG associated with the temporal acceleration signal of rolling bearing vibration by using the method mentioned above. However, the graph cannot be directly analyzed numerically. Therefore, we gleaned the VG features from previous studies as much as possible. The graph density [53], degree distribution [54] and graph index complexity are selected to be global VG features, and the Gaussian Markov random fields (GMRF) are used to produce the local VG features.

### 4.1. Candidate VG Features

The candidate VG feature set is composed of the global VG feature subset and the local VG feature subset. The global VG features come from graph theory. The three global VG features used in this paper include the graph density, standard deviation of node degree and graph index complexity, and the 12 local VG features used in this paper refer to the 12 GMRF parameters from the five-order traditional GMRF analysis of VGAM.

#### 4.1.1. Global VG Features

(1)VG density:

The VG density (VGD) reflects the size of graph or its adjacent matrix. Let *G* denote the visibility graph transformed from vibration signals. The density $G_{d}$ of *G* is defined as a ratio of the number of edges to the number of possible edges in the graph. Let Ddenote the VGAM, the VG density can also be represented by the VGAM, that is:(3)$$G_{d}=\frac{\sum_{i,j}m_{i,j}}{N(N-1)}$$

If the signal envelope is composed of much longer concave EC than convex EC, more signals can be connected, and thus, the VG density is larger, otherwise, the VG density is lower.

(2)VG complexity:

The VG complexity (VGC) is to signify the global complexity of the complex graph structure and adjacent matrix. Let $\lambda_{max}$ be the maximum eigenvalue of the adjacency matrix of VG; VGC is defined as follows [55]:(4)VGC=4κ(1−κ)
where:(5)$$\kappa =\frac{\lambda_{max}-2cos\left(\pi /\left(N+1\right)\right)}{N-1-2cos(\pi /(N+1))}$$
in which $2cos\left(\pi /\left(N+1\right)\right)\leq \lambda_{max}\leq n-1$ [56]. It has been proven that that the graph complexity strongly depends on the number of edges. Unlike the VG density, a medium number of edges alone already guarantees a high VGC [55]. Further, the VGC index has been also used to characterize electroencephalograms [38]. Therefore, we can use the VGC to distinguish the graph with a medium number of edges, which is determined by the distribution of convex and concave EC, as described in Section 4.1.1.

(3)VG degree:

The VG degree is the number of connections or edges the signal *i* has to the other nodes. For the vibration signals, a smaller or larger value (compared with other points) makes them visible or invisible by many other points, which in turn causes their corresponding nodes to be or not to be the hot spot of the graph with many connections. The degree of signal *i* can be formulated as:(6)$$D\left(i\right)=\sum_{j}m_{i,j}$$

However, the VG degree is array data, so feature preprocessing is thus needed [57], and the mean value and standard deviation of the VG degree are suggested in this paper. Because the mean value of the VG degree is N−1-times VG density $G_{d}$, only the standard deviation of the VG degree is adopted in the experiments.

#### 4.1.2. Local VG Features

The local VG features are extracted by considering the binary matrix as the gray-level matrix of binary images. Gaussian Markov random fields (GMRF) have been shown to perform better than other models in both the classification and segmentation of textured images [52,58]; therefore, we use the GMRF to extract the local VG features. In the GMRF model, the $d_{i,j}$ in VGAM *D* is represented by y(s).
(7)y(s),s∈Ω,Ω=s=(i,j):0≤i,j≤N−1
for the N×N VGAM *D*. The Markov random field models are described by the conditional probability $P(y\left(s\right)|y\left(\eta_{s}^{n}\right))$, and $\eta_{s}^{n}=s+r,r\in \eta^{n}$ is an *n*-th-order symmetric neighbor set of site *s*. The first- to fifth-order neighbor MRF relationships are depicted in Figure 4.

The GMRF model assumes that the y(s) obey the following equation:(8)$$y\left(s\right)=\sum_{r\in \eta_{s}}\theta_{r}y\left(s+r\right)+e\left(s\right)$$
where the $\eta_{s}$ is a neighbor set dependent on the order and type of model used and $\theta_{r}$ is the GMRF parameter for neighbor *r* and characterizes the local properties of VGAM. The parameter set $\theta_{r}$ satisfies:(9)$$\theta_{r}=\theta_{-r},r\in \eta_{s}$$
e(s) is a stationary Gaussian noise sequence defined by:(10)$$e\left(s\right)=\frac{1}{\sqrt{2\pi \sigma^{2}}}exp\left(-\frac{y{\left(s\right)}^{2}}{2\sigma^{2}}\right)$$

Equation (Equation 8) can be also represented by the matrix:(11)$$y\left(s\right)=\theta^{T}Q_{s}+e_{s}$$
where θ is a vector comprised of $\theta_{r}$ and $Q_{s}$ is a vector defined by:(12)$$Q_{s}={\left[y_{s+r1}+y\left(s-r1\right),\ldots ,y\left(s+rn\right)+y\left(s-rn\right)\right]}^{T}$$

Then, the vector θ and variance σ can be estimated using the least-squares (LSQR) approach presented in the following equations:(13)$$\hat{\theta }={\left[\sum_{s\in \Omega }Q_{s}Q_{s}^{T}\right]}^{-1}\left[\sum_{s\in \Omega }Q_{s}y\left(s\right)\right]$$
(14)$$\hat{\sigma }=\frac{1}{L^{2}}\sum_{s\in \Omega }\left[y\left(s\right)-{\hat{\theta }}^{T}Q_{s}\right]$$
where *L* denotes the length of y(s). In this paper, we adopt the fifth-order GMRF, which produces 12 GMRF parameters $\hat{\theta }$.

In summary, three global and 12 local VG features are extracted. However, it is not certain that all the VG features are necessary for fault diagnosis because some VG features may reflect similar properties of vibration signals; therefore, we should select the necessary features that greatly contribute to the fault diagnosis.

### 4.2. Diagnosis Performance-Based VG Feature Selection

In this section, our goal is to select the necessary features that greatly contribute to the fault class diagnosis. The SFS algorithm [59] is applied to the VG feature selection. The algorithm starts with an empty set and adds one VG feature for the first step, which gives the highest value for the objective function. From the second step onwards, the remaining VG features are added individually to the current subset, and the new subset is evaluated.

A multiple-classifier including support vector machine (SVM), k-nearest neighbor (KNN) and artificial neural network (ANN) classifier are applied to detect the bearing fault classes and produce the significance of each VG feature with regard to the updated feature subset. The diagnosis accuracy (ACC) is selected as the criteria to judge the classification performance. The accuracy (ACC) or its complement, i.e., the error rate, is the most easily understandable index. Let ξ denote the ACC, *E* denote the number of testing samples and $E_{r}$ denote the number of testing samples that are classified correctly; ξ is equal to $E_{r}/E$.

The significance of each VG feature $f_{i}$ with regard to the updated feature subset $F_{c}$ is the difference between the ACC ξ obtained by using VG feature subsets $F_{c}$ and $F_{c}\cup f_{i}$. The individual VG feature with the most significance is permanently included in the subset $F_{c}$ in each step.

## 5. Experiments

### 5.1. Data and Experiments’ Description

The two datasets used in this paper come from the Case Western Reserve University (CWRU) Bearing Data Centre and Intelligent Maintenance Systems (IMS) bearing run-to-failure dataset because the two datasets have been widely used in previous studies [4,50,60,61,62]. In the IMS dataset, 12 bearings in the experimental setup were tested in the same conditions of speed and load. Despite that, only four bearings were broken. The total number of records is 5394. Each record is formed by 20,480 signals. Because of the overlap between normal-degradation zones and degradation-faulting zones, the number of records is reduced to 3000. Three kinds of fault type including normal bearings, outer race fault and inner race fault are selected in the experiments. More information about the data structure can be found on the website (http://data-acoustics.com/measurements/bearing-faults/bearing-4/).

The CWRU dataset consists of time series of the vibration acceleration value (vibration signal) measured from a sensor mounted on the bearing housing at the drive end (DE), fan end (FE) and base of the induction motor. The vibration signals of the rolling bearings were obtained from different states—(1) normal operating conditions (NM); (2) inner race fault (IR); (3) rolling ball fault (RB); (4) outer race fault (OR)—and recorded for motor loads of 0–3 horsepower (motor speeds of 1797–1720 RPM). Fault levels ranging from 0.007 inches in diameter to 0.0210 inches in diameter were introduced separately for each fault class. More information about the data structure can be found on the website (http://csegroups.case.edu/bearingdatacenterpages/download-data-file). The dataset used in the experiment is acquired by the sensor at the DE when the bearing at the DE has faults because the sensor at the FE can detect the faults at the DE with less confidence, and the sampling frequency is 12 kHz.

In the experiment, 1000 signals are considered as one sample. Table 1 and Table 2 list the machine condition classes defined for the experiments. Table 1 lists machine condition classes of IMS data, while Table 2 lists machine condition classes of CWRU data. Note that the number of classes is much more extensive than that found usually in work related to the CWRU data. We tried to distinguish among the different bearing fault locations and the severity of the fault (i.e., the diameter of the artificially-drilled hole into the material). Even different loads for the same fault were considered as separate classes, which constitutes the most challenging classification problem. There are 21 classes to be distinguished, and the details about all classes are shown in Table 2.

In the experiments, we used the Sklearn [63], which is a Python package to realize the proposed classification algorithm. The statistical features including time-domain and frequency features and wavelet packet features are the typically used inputs of classifiers [50,64]; therefore, we compare the performance of the proposed VG features and 13 statistical features in this section. The 13 statistical features used in this paper include root mean square (RMS), square root of the amplitude (SRA), kurtosis value (KV), skewness value (SV), peak-to-peak value (PPV), crest factor (CF), impulse factor (IF), margin factor (MF), shape factor (SF), kurtosis factor (KF), frequency center (FC), RMS frequency (RMSF) and root variance frequency (RVF). The procedure proposed in [11] is used to produce the wavelet packet features. The mother wavelet is Daubechies 4, and refining is done down to the fourth decomposition level. Sixteen ($2^{4}$) wavelet features are obtained finally. The n−th,(n=1,2,…,16) wavelet package feature is the respective percentages of the energy of each leaf.

### 5.2. VGAM Construction

First, subjecting the bearing fault data to the visibility approach defined in Section 3, the VGAM, which may represent the characteristics of vibration signals, is obtained finally. The examples of VGAM converted from vibration signals are shown in Figure 5. It is clear that the temporal distribution of 0-1 (black-white) varies for different fault types. However, the VGAM is so complicated that it is necessary to extract features to describe the VGAM. The VG features for representative machine conditions in the section are calculated. Fifteen VG features are obtained finally.

### 5.3. Results and Discussion

In this section, the optimal feature subset is solved based on the proposed VG feature selection algorithm described in Section 4.2. The linear SVM (LSVM), 1-NN and MLP were used in the experiments, and the number of hidden layers was equal to the number of features for MLP. All classifiers always used 10-fold cross-validation to estimate the diagnosis ACC. Figure 6 and Figure 7 show the diagnosis ACC of the three classifiers during the process of feature selection using IMS and CWRU data, respectively. Based on the obtained ACC with different number of VG features, the optimal number of features can be produced.

With respect to the experiments using the IMS dataset, the ACC of the 1-NN and LSVM classifiers is depicted in Figure 6a,b. The statistical feature model performed better than the other two feature models. However, the proposed VG feature model had lower ACC than the other two feature models. The ACC (99.33% and 99%) was also acceptable for bearing fault diagnosis. The ACC of the MLP classifier is depicted in Figure 6c. The proposed VG feature model performed better than the wavelet package feature model because the ACC of fault diagnosis was 100%.

With respect to the the experiments using the CWRU dataset, the ACC of the 1-NN classifier is depicted in Figure 7a. The statistical feature model performed better than the other two feature models when the number of selected features was less or equal to two. However, the ACC was unacceptable for bearing fault diagnosis. The proposed VG feature model performed better than the other two feature models when the number of selected features was greater than two. The maximum ACC of 1-NN using the VG feature model (98.0%) was higher than the statistical features (83.3%) and the wavelet package features (94.5%). The corresponding optimal number of features was 11, 9 and 10 for the VG, statistical and wavelet package feature model.

The ACC of the LSVM classifier is depicted in Figure 7b. The proposed VG feature model performed better than the other two feature models when the number of selected features ranged from 1–15. The maximum ACC of LSVM using the VG feature model (98.6%) was also higher than the statistical features (85.4%) and the wavelet package features (96.6%). The corresponding optimal number of features was 15, 13 and 15 for the VG, statistical and wavelet package feature model.

The ACC of the MLP classifier is depicted in Figure 7c. The proposed VG feature model performed better than the other two feature models when the number of selected features ranged from 1–15. The maximum ACC of LSVM using the VG feature model (98.9%) was also higher than the statistical feature model (86.7%) and the wavelet package feature model (96.4%). The corresponding optimal number of features was 11, 9 and 14 for the VG, statistical and wavelet package feature model.

All in all, the optimal VG feature subset was obtained by comparing the performance of three classifiers. The optimal VG feature subset was composed of three global VG features and eight local VG features based on the results of the 1-NN and MLP classifiers. This fact shows that both the global and local VG features should be extracted for bearing fault diagnosis. The diagnosis ACC was so high that the VG features can be considered as the input of classifiers. Because the working condition and the fault level were also considered in the experiments, the VG features can be used to recognize the fault class and the corresponding fault level of rolling bearings under the same or variable working conditions. The three classifiers using the VG feature model performed better than the statistical feature model and wavelet package feature model, which are often used as the input of classifiers. In the three classifiers, the MLP using the VG feature model performed best.

## 6. Conclusions and Extension

The paper proposed a framework to extract VG features from the vibration signals of rolling bearings, which are used widely in industrial systems. Firstly, the VG approach is used to convert the normalized vibration signals into a binary matrix. It is a successful attempt to analyze vibration signals even though the VG approach has been used to analyze data in other areas. The approximate homogeneity between the VG of signals and the signal envelope is analyzed based on Jensen’s inequality. The VG feature pool, which is comprised of 15 VG features, has been extracted to describe the VGAM structure.

Secondly, the sequential feature selection (SFS) method is applied to select the optimal VG feature subset. The classifiers including LSVM, 1-NN and MLP are applied to validate the VG feature model based on the CWRU data. The results show that the VG feature model can be used to analyze the vibration signals because the selected VG features can be taken as the input of the classifiers in order to recognize the fault precisely. The VG feature model can be used to recognize the fault class of rolling bearings under the variable working conditions, which means that our model can predict the localization of the fault and its severity without knowing the load and rotation speed clearly.

Finally, the advantage of the VG feature model is verified by comparison with the statistical feature model and the wavelet package feature model, and the VG feature model performs better than the other two feature models. While the results generated using the proposed method were based on limited test data, they offer useful insights into fault diagnosis. The VG features are novel and can be selected by engineers to detect bearing faults.

In the future study, we will validate the VG features by using vibration signals of other rolling bearings. The proposed method may also serve as an analytical method to detect the faults of other machines such as gear boxes using the combination of the VG feature model and other features in the future. Considering the real-time condition monitoring of rolling bearings, a resampling method of vibration signals will be proposed to reduce the computing complexity of the visibility graph in our future works.

## Figures and Tables

**Figure 1 materials-11-02262-f001:**
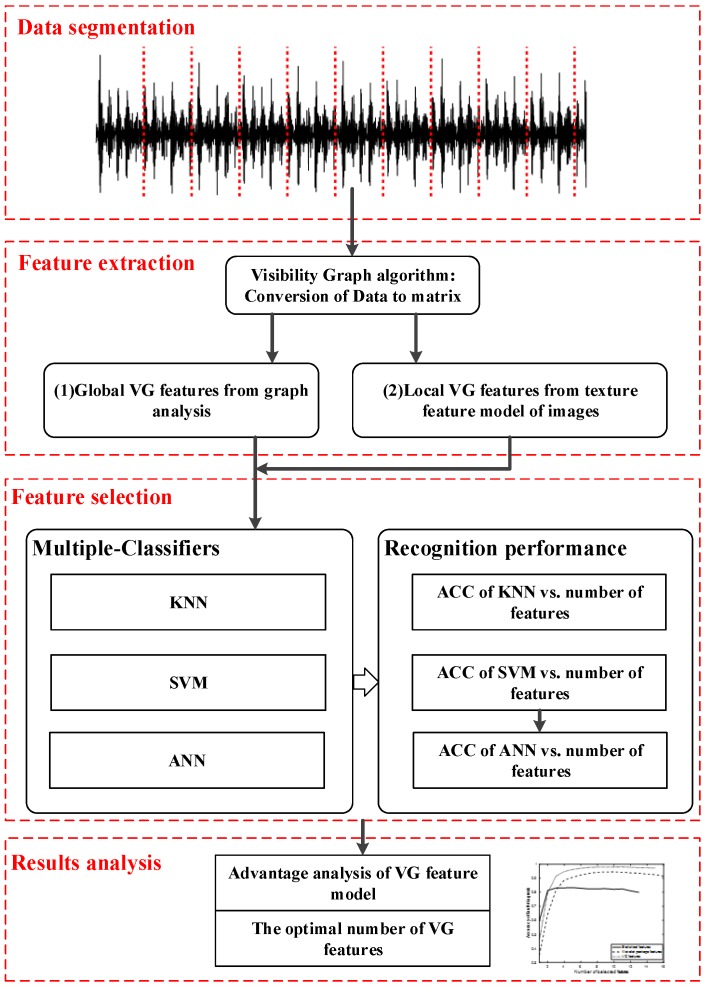
Framework of the VG feature model for bearing fault classification.

**Figure 2 materials-11-02262-f002:**
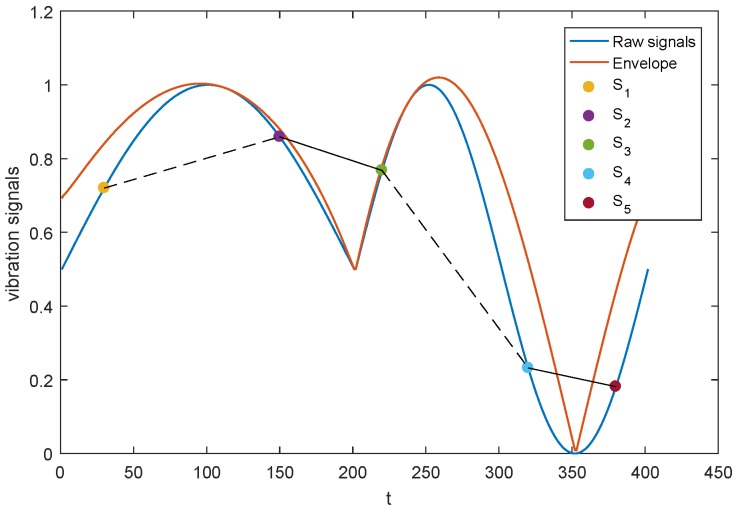
Relationship between the signal envelope and the visibility graph.

**Figure 3 materials-11-02262-f003:**
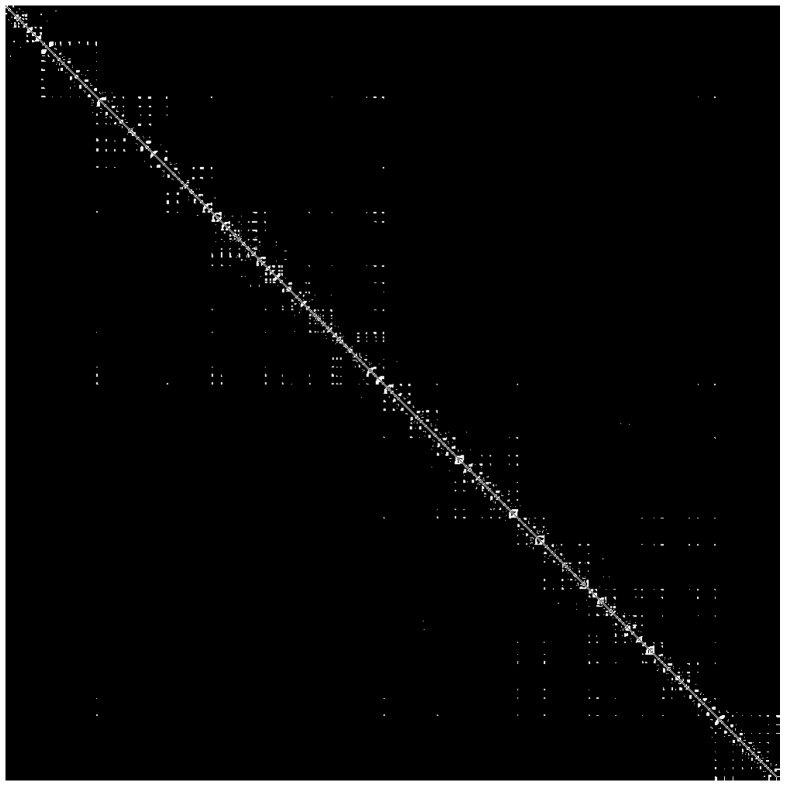
The binary image of the adjacent matrix of the visibility graph (VGAM).

**Figure 4 materials-11-02262-f004:**
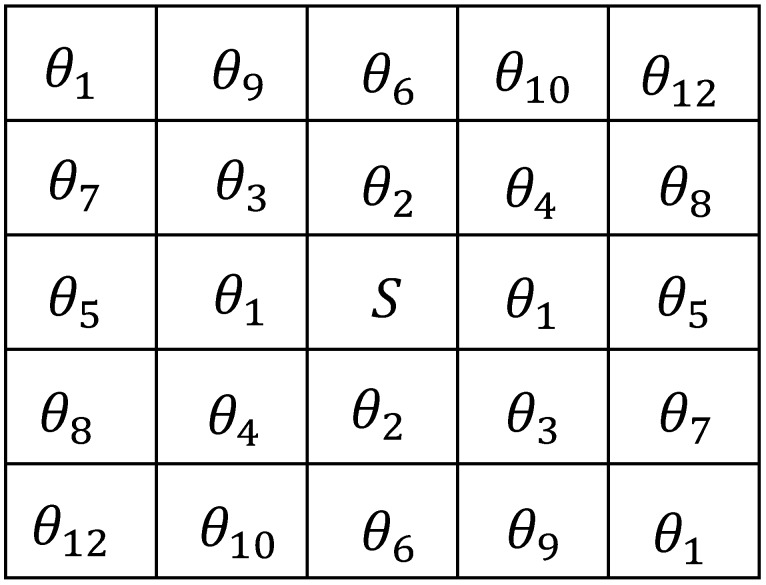
The parameters of the Markov random fields (MRF) model for the first-order neighbor to the fifth-order neighbor.

**Figure 5 materials-11-02262-f005:**
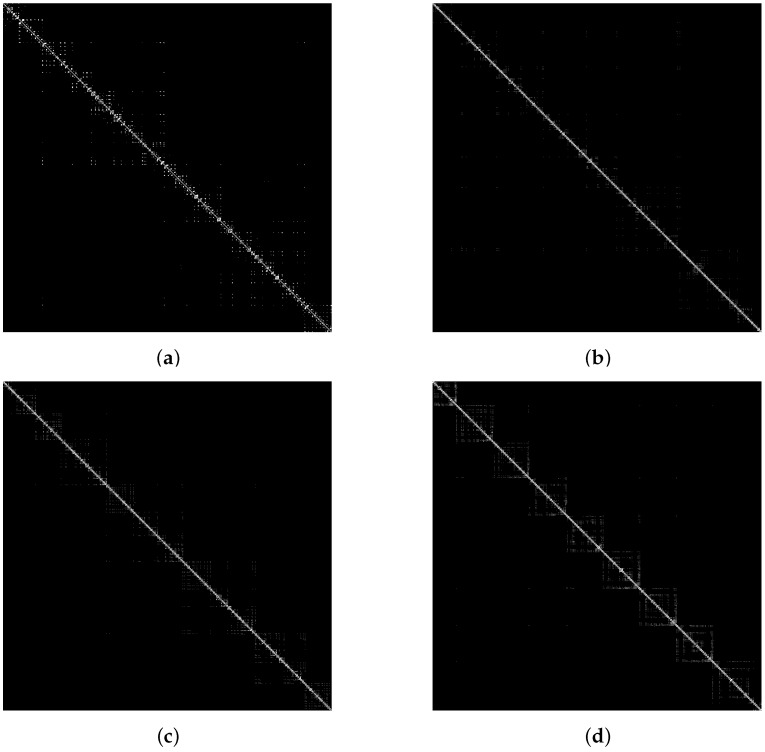
VGAM from vibration signals. (**a**) NM; (**b**) RB; (**c**) IR; (**d**) OR.

**Figure 6 materials-11-02262-f006:**
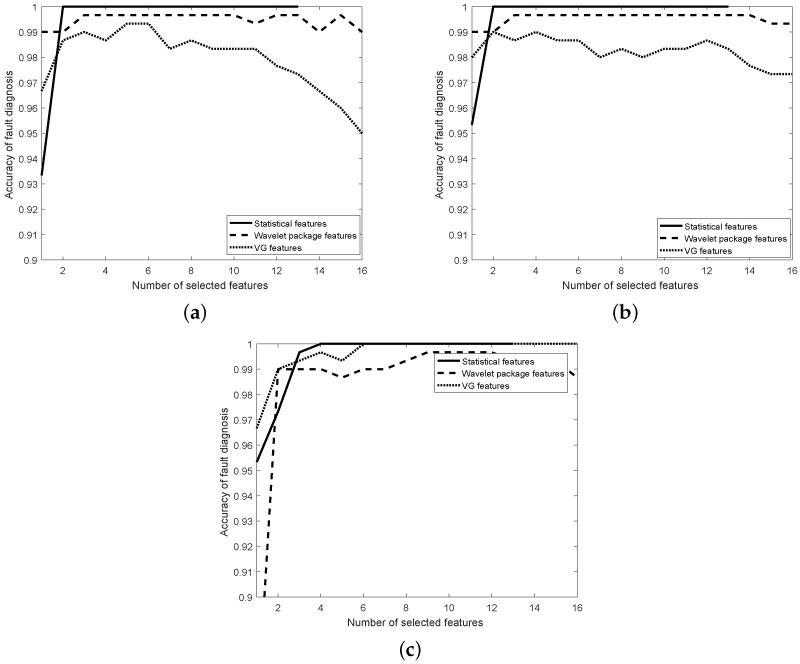
Estimated accuracy during feature selection for all features using IMS data. (**a**) KNN; (**b**) LSVM; (**c**) MLP.

**Figure 7 materials-11-02262-f007:**
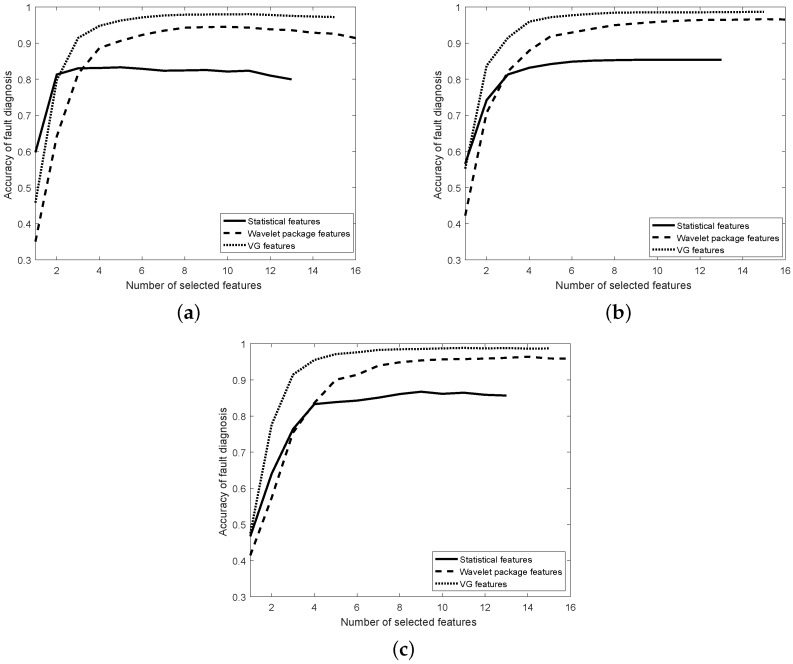
Estimated accuracy during feature selection for all features using the CWRU data. (**a**) KNN; (**b**) LSVM; (**c**) MLP.

**Table 1 materials-11-02262-t001:** Class distribution and description of the Intelligent Maintenance Systems (IMS) dataset.

Class	Name	Samples	Distribution	Data Description
1	NM	100	33.33%	normal bearings
2	OR	100	33.33%	Outer race fault
3	IR	100	33.33%	Inner race fault

**Table 2 materials-11-02262-t002:** Class distribution and description of the Case Western Reserve University (CWRU) dataset. RB, rolling ball.

Class	Name	Samples	Distribution	Data Description
1	NM_0	100	2.083%	NM load = 0
2	NM_1	100	2.083%	NM load = 1
3	NM_2	100	2.083%	NM load = 2
4	NM_3	100	2.083%	NM load = 3
5	IR007	400	8.333%	IR fault level = 0.007
6	IR014_0	100	2.083%	IR fault level = 0.014 load = 0
7	IR014_1	100	2.083%	IR fault level = 0.014 load = 1
8	IR014_2	100	2.083%	IR fault level = 0.014 load = 2
9	IR014_3	100	2.083%	IR fault level = 0.014 load = 3
10	IR021	400	8.333%	IR fault level = 0.021
11	OR007	400	8.333%	OR fault level = 0.007
12	OR014	400	8.333%	OR fault level = 0.014
13	OR021	400	8.333%	OR fault level = 0.021
14	RB007	400	8.333%	RB fault level = 0.007
15	RB014	400	8.333%	RB fault level = 0.014
16	RB021_0	100	2.083%	RB fault level = 0.021 load = 0
17	RB021_1	100	2.083%	RB fault level = 0.021 load = 1
18	RB021_2	100	2.083%	RB fault level = 0.021 load = 2
19	RB021_3	100	2.083%	RB fault level = 0.021 load = 3
20	IR028	400	8.333%	IR fault level = 0.028
21	RB028	400	8.333%	RB fault level = 0.028

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
