# Peer review of "Visibility Graph Feature Model of Vibration Signals: A Novel Bearing Fault Diagnosis Approach"

_materials, 2018, doi:10.3390/ma11112262_

Reviewer 1 Report

The paper presents a new method for rolling bearings diagnosis, called visibility graph (VG) feature model. This method supposes the transformation of the acquired acceleration signals in a binary matrix. The authors prove that visibility graph can reflect the characteristics of signal envelope. Subsequently, the global and local features of the VG are extracted for bearing fault diagnosis.

The proposed model is original, and I appreciate its complexity and the fact that the authors underlined its limitations:

- The results refer to a limited data, available from rolling element bearing diagnostics database of Case Western Reserve University, and may not always be applicable to all rolling bearing diagnosis.

- The model is not validated for other rolling bearings.

- The method is time consuming and must be combined in the future with a resampling method of vibration signals.

Taking into account the above mentioned aspects, I suggest to the authors some improvements:

1. The entire article is written in a confusing mode, as it is asserted that the model is dedicated for bearing diagnosis and in the same time the authors express their concern upon the fact that the studied database of signals, coming from a single bearing with singular geometrical parameters, may be a limitation of the method. I advise the authors to analyze just one single diagnosing signal coming from another faulty rolling bearing if available, and to present the results in the comparative way they did using also statistical model and wavelet package. In such a way, it will be clear if the VG model is suited as a diagnosis tool, or it must be considered just a fault classification tool of an existing database of acquired signals.

2. In Conclusion section it is claimed that the VG feature model can be used to recognize the fault class of rolling bearing under the variable working condition. From here the reader understands that the model can predict not only the faulty state, but the localization of the fault and its severity. This is normal when you use an existing database, but can your model predict the fault frequency of equipment? Please clarify this aspect too.

3. In subsection 5.3, line 236, the authors indicates the subsection 5.2 for the description of used algorithm, but 5.2 does not present any algorithm.

4. There is a conflict between statements in Conclusions section, lines 283-286. At first it is mentioned that the method is an useful insight into rolling bearing diagnosis and may not always be applicable for all rolling bearings, and after you specify that VG features are novel and can be selected by engineers to detect bearing faults. Please amend this aspect too.

5. In subsection 5.1., line 210, a reference to Table III is done instead of Table 1.

Author Response

1. The entire article is written in a confusing mode, as it is asserted that the model is dedicated for bearing diagnosis and in the same time the authors express their concern upon the fact that the studied database of signals, coming from a single bearing with singular geometrical parameters, may be a limitation of the method. I advise the authors to analyze just one single diagnosing signal coming from another faulty rolling bearing if available, and to present the results in the comparative way they did using also statistical model and wavelet package. In such a way, it will be clear if the VG model is suited as a diagnosis tool, or it must be considered just a fault classification tool of an existing database of acquired signals.

Answer: Thanks. We have add an extra experiment in the revised manuscript using the IMS data set. The comparison results show that the proposed VG feature model also has good performance (The diagnosis accuracy is above 99%) although the statistical feature model performs better than our model. Based on the two experiments, the proposed model is more robust than the other two models.

2. In Conclusion section it is claimed that the VG feature model can be used to recognize the fault class of rolling bearing under the variable working condition. From here the reader understands that the model can predict not only the faulty state, but the localization of the fault and its severity. This is normal when you use an existing database, but can your model predict the fault frequency of equipment? Please clarify this aspect too.

Answer: Thanks. The experiments show that the VG feature model can be used to recognize the fault class of rolling bearing under the variable working condition. The working condition refers to the load and rotation speed. However, it is not validated that the model can predict the fault frequency of equipment. Usually fault frequency of equipment changes along with the various rotating speeds. Therefore, our model can predict the localization of the fault and its severity without knowing the load and rotation speed clearly.

3. In subsection 5.3, line 236, the authors indicates the subsection 5.2 for the description of used algorithm, but 5.2 does not present any algorithm.

Answer: Thanks, we have revised the notation, the reference should be subsection 4.2

4. There is a conflict between statements in Conclusions section, lines 283-286. At first it is mentioned that the method is an useful insight into rolling bearing diagnosis and may not always be applicable for all rolling bearings, and after you specify that VG features are novel and can be selected by engineers to detect bearing faults. Please amend this aspect too.

Answer: Thanks, it is clear that the VG model is suited as a diagnosis tool because we have used two datasets to verify our model

5. In subsection 5.1., line 210, a reference to Table III is done instead of Table 1.

Answer: Thanks, we have revised the reference to table in the manuscript.

Reviewer 2 Report

Do changes according to mandatory changes in file attached.

Author Response

Conclusion. Give them as bullets, one per each higthligths.

Answer: Thanks for your suggestion. We have used “firstly, secondly, finally” to describe the highlights of this paper clearly.

Ref 39 is out of date, please eliminate.

Answer: Thanks for your suggestion. We have deleted it.

Figure 5: please improve the explanation. Now they are black images without a real meaning.

Answer: Thanks for your suggestion. In the text above the figure, we have added short explanation of figure 5 in the revised manuscript: “It is clear that the temporal distribution of 0-1(black-white) varies for different fault types. However the VGAM is so complicated that it is necessary to extract features to describe the VGAM”. The figure is composed of black–white (0-1 adjacent matrix)”. It is also clear that the white points are much less than the black points. The fact also motivates us to improve the computing efficiency of the proposed model in the future.

Material control must be of paramount important; you can add some information about that (this is a materials journal). For instance, in Materials and Manufacturing Processes 26 (8), 997-1003 the effect on surface finishing in relation with part quality is mentioned, or in The International Journal of Advanced Manufacturing Technology 62 (5-8), 505-515 the steel composition on final mechanical performance of materials.

Answer: Thanks for your suggestion. In the 1st paragraph of introduction, we have presented that “ the material control in the quality design process\cite{lopez2011five}\cite{fernandez2012behavior} and the fault diagnosis of rolling bearings in the operation process have attracted considerable attention of engineers and scholars in order to reduce the system breakdown”

Modeling: this is key as well, because early detection is the basic of bearing performance and behavior. The paper promotes a new view of monitoring, perhaps related with real industrial needs. MDPI also promotes the idea, see https://doi.org/10.3390/machines5020015. ANN can help, in recent works the idea was tackled with care, see Journal of manufacturing systems 48, 108-121

Answer: Thanks for your suggestion. In the 4th paragraph of introduction, we have presented that “ The artificial intelligence (AI) has been developed and applied to reduce the breakdowns of manufacturing systems\cite{urbikain2017reliable}\cite{bustillo2018smart}.” 

Round  2

Reviewer 1 Report

The authors took into account all my suggestions and I congratulate them for the good work.

Reviewer 2 Report

Figure 3: please improve including more information: points, labels, etc.

Conclusions: give them as bullets, one per each highlight.